# Nanoparticles in Plants: Uptake, Transport and Physiological Activity in Leaf and Root

**DOI:** 10.3390/ma16083097

**Published:** 2023-04-14

**Authors:** Xueran Wang, Hongguo Xie, Pei Wang, Heng Yin

**Affiliations:** 1College of Transportation Engineering, Dalian Maritime University, Dalian 116026, China; wangxueran@dicp.ac.cn (X.W.); peterwp@dlmu.edu.cn (P.W.); 2Dalian Engineering Research Center for Carbohydrate Agricultural Preparations, Dalian Technology Innovation Center for Green Agriculture, Liaoning Provincial Key Laboratory of Carbohydrates, Dalian Institute of Chemical Physics, Chinese Academy of Sciences, Dalian 116023, China

**Keywords:** nanoparticles, uptake, transport, physiological activity

## Abstract

Due to their unique characteristics, nanoparticles are increasingly used in agricultural production through foliage spraying and soil application. The use of nanoparticles can improve the efficiency of agricultural chemicals and reduce the pollution caused by the use of agricultural chemicals. However, introducing nanoparticles into agricultural production may pose risks to the environment, food and even human health. Therefore, it is crucial to pay attention to the absorption migration, and transformation in crops, and to the interaction with higher plants and plant toxicity of nanoparticles in agriculture. Research shows that nanoparticles can be absorbed by plants and have an impact on plant physiological activities, but the absorption and transport mechanism of nanoparticles is still unclear. This paper summarizes the research progress of the absorption and transportation of nanoparticles in plants, especially the effect of size, surface charge and chemical composition of nanoparticle on the absorption and transportation in leaf and root through different ways. This paper also reviews the impact of nanoparticles on plant physiological activity. The content of the paper is helpful to guide the rational application of nanoparticles in agricultural production and ensure the sustainability of nanoparticles in agricultural production.

## 1. Introduction

As the global population expands and dietary habits evolve, there has been a noticeable surge in the need for food. In response, nanotechnology has been employed to optimize the yield and quality of agricultural commodities [1]. Nanoparticles possess unique properties in comparison to bulk material and have the potential to offer an innovative solution for nutrient deficiency in crops, promote plant growth and development, and inhibit plant pathogens.

Nanoparticles have a wide range of applications in agriculture, including modifying plant genetics, improving crop growth and development, and controlling the release of agrochemicals [2,3,4]. Nanoparticles facilitate site-targeted delivery and the controlled release of agrochemicals and various macromolecules needed for plant growth, ensuring efficient utilization and reduced exposure for eco-protection [5]. Furthermore, nanoparticle-mediated plant transformation technology allows for genetic modification in plants more effectively than traditional methods. Nanoparticles can carry foreign substances into plant cells due to their small size, while protecting foreign substances from degradation [6]. Lastly, nanoparticles provide a novel method for crop protection against specific agricultural problems [7]. Nonetheless, it has been identified that some nanoparticles may negatively impact plant growth and cause phytotoxicity [8]. Therefore, it is crucial to know which parameters (chemical, biological and environmental) will influence the fate of a nanoparticle after it is applied to plants.

It is generally believed that nanoparticles can be uptaken and transported by plants, and their uptake and transport of nanoparticles in plants depend on various factors such as particle size, surface charge, concentration, exposure time and plant species. Nanoparticles can enter the plant system through several pathways such as stomata, root hairs and cracks on the leaf surface. Once inside the plant, nanoparticles can move through the plant system by diffusion, bulk flow and phloem loading. The transport of nanoparticles can be influenced by various factors such as the size and shape of nanoparticles, their surface properties, the pH of the solution and the presence of other ions or compounds in the solution. Previous research has mainly applied nanoparticles to plants by method of leaf spraying, root application, branch injection and seed treatment. Some research confirmed the uptake of nanoparticles by plants [9,10]. However, direct proof of transport of nanoparticles in plants is limited [11]. Understanding the uptake and transport behavior of agricultural nanoparticles in plants is crucial for designing optimal nanoparticles for agricultural use [2,4]. At the same time, understanding the mechanism behind their action and bioaccumulation of agricultural nanoparticles in plants can help to clarify the biological safety of nanoparticles and provide guidance for the safe use of agricultural nanoparticles [9]. Although studies have shown that nanoparticles can remain in edible parts of plants and be transferred to plant produce consumers through the food chain, affecting human health and food security, the transfer of nanoparticles into animals through plants and their impact on plant produce consumers are not covered in this review. This review presents insights into the uptake and transport behavior of agricultural nanoparticles in plants, as well as their effects on the physiological activities of occupational plants, especially the effect of particle size, surface charge and plant species on uptake and transport of nanoparticles.

## 2. Uptake of Nanoparticles in Plant Leaves

### 2.1. Pathways for Foliar Uptake

During agricultural applications, nanoparticles are typically sprayed onto the leaf surface when they deposit, and are subsequently absorbed by plants through either the cuticle or stomata on leaf surface. The waxy cuticle of leaf epidermis is mainly composed of wax, cutin and pectin. It protects plant leaves from loss of water during plant growth, and act as a primary natural barrier to prevent nanoparticles from entering the leaves [12,13]. However, there are two different channels on the surface of waxy stratum corneum [14]; one channel is hydrophilic, the other is lipophilic. The diameters of both hydrophilic and lipophilic channels vary from 0.6 nm to 4.8 nm [15]. The hydrophilic channels allow hydrophilic nanoparticles less than 4.8 nm in diameter to diffuse [16]. The lipophilic channels in the surface of cuticle allow lipophilic nanoparticles to be absorbed by leaves through diffusion and infiltration [17]. Recently, Hu et al. proved that carbon dots with a size of less than 2 nm could enter cotton leaves through the cuticular pathway by using confocal fluorescence microscopy with high spatial and temporal resolution [18]. However, due to the limited size of the pore channels in the cuticle, the absorption of nanoparticles by plants through the epidermis is limited. Some research reports indicated that nanoparticles could accumulate in the leaf epidermis and vascular tissue after being applied on the leaf surface. Meanwhile, a large number of studies observed that nanoparticles could be translocated to other tissues of plants. The researchers proposed that nanoparticles could be absorbed by another way, namely the stomatal pathway (Figure 1).

The stomata on leaf surface play a crucial role in regulation water and gas exchange in plants. The size of stomata is generally 10–100 μm. The size and density of stomata vary among different plant species. Due to the unique geometric construction and physiological function of stomata, the actual size exclusion limit (SEL) of stomatal aperture for nanoparticle diffusion is still unknown [19].

### 2.2. Key Factors That Affecting the Uptake of Nanoparticles via Leaves

Current research has shown that the uptake of nanoparticles in plant depends on the nature of the nanoparticles, plant species and the environment. The properties of nanoparticles such as particle size, chemical composition, surface charge and surface modification can affect their absorption behavior in plant leaves. Due to the size exclusion limit of NPs in the blade absorption pathway, the particle size of nanoparticles has become one of the most important factors in the study of absorption of NPs in the blade [20].

#### 2.2.1. Effect of Size and Chemical Composition

The effect of size on the uptake of metal-based nanoparticles has been studied extensively. Based on the studies, it has been found that metal-based nanoparticles with a diameter of less than 50 nm could enter plant leaves via the stomatal pathway [19]. With an increase in particle size, the absorption ability of leaves to nanoparticles decreased. Many of studies have identified the foliar uptake of nanoparticles. For instance, Zhu et al. applied fluorescein isothiocyanate (FITC) labeled ZnO nanoparticles (30 nm) to wheat leaves. By confocal microscopy, they found that ZnO nanoparticles mainly passed through the epidermis of wheat leaves by the stomata pathway, and then accumulated in chloroplasts [21]. They also investigated the effect of stomatal opening and closing on the absorption of ZnO NPs. It was proven that with the decrease in stomatal diameter, the concentration of zinc in chloroplast and cytoplasm of wheat leaf cells decreased by 33.2% and 8.3%, respectively, with the decrease in stomatal diameter. Avelian et al. used gold nanoparticles with a different diameter (3, 10, 50 nm) modified with coatings to act on wheat leaves. They found that the coated gold nanoparticles of all sizes could be absorbed by wheat (*Triticum aestivum* cv. *cumberland*) leaves [14]. At same time, their results indicated that the uptake of gold nanoparticles occurred via disruption and/or diffusion through both cuticle layer and stomata. In another study, researchers confirmed that MgO nanoparticles with a size of 27–35 nm were absorbed in watermelon (*Citrullus lanatus*) leaves by transmission electron microscopy (TEM). In addition to metal-based nanoparticles, researchers also found that nanoparticles made of silica, polymer and natural materials could be absorbed by plant leaves. Their research results indicated that the type of nanoparticle had a great effect on the critical size that could be absorbed by plant leaves. By using TEM, researchers confirmed that SiO_2_ with size of 54 nm could enter the leaves of model plant *Arabidopsis thaliana* through the stomatal pathway [22]. Zhao et al. treated cucumber leaves with FITC-labeled mesoporous silica nanoparticles (MSNs) (particle size 200–300 nm), and found that FITC-labeled MSNs could be absorbed by the leaves. Researchers found that polystyrene nanoplastic with size of 93.6 nm could cross the barrier on lettuce leaf and enter the phloem in plants via TEM [23]. Recently, researchers found that chitosan-based silicon nanoparticles with the average size of 166 nm were absorbed by rice (Zhenghan 10) leaves and were distributed in the leaf cells [24].

#### 2.2.2. Effect of Shape and Surface Charge

In addition to the size of nanoparticles, the entry of nanoparticles into plant mesophyll tissue also depends on the shape and charge. Nanoparticles with different shapes have different interfacial properties that lead to changes in the surface area of nanoparticles and the contact angle with the plant surface, ultimately affecting the absorption of NP [25,26]. Zhang et al. compared rod-shaped gold nanoparticles with spherical nanoparticles and found that rod-shaped nanoparticles were more easily absorbed and internalized by *Arabidopsis* leaves under the condition of similar particle size [27]. Both positively- and negatively-charged nanoparticles can be absorbed by plant leaves. Previous studies have evaluated the effect of surface charge on the absorption of surface charge on graphene quantum dots (GQDs) on maize leaves [28] and on the adsorption of ZnO nanoparticles on wheat leaves [26]. The results showed that both positively-charged NH_2_-GQDs (13 nm) and negatively-charged OH-GQDs (14 nm) can be absorbed by maize leaves via stomata. FITC-labeled F-P-ZnO NPs (40 nm) with a positive charge and F-N-ZnO NPs (40 nm) with a negative charge were confirmed by confocal microscopy to gather at the stomata on the surface of wheat leaves. At the same time, it showed that the adsorption of positively-charged nanoparticles in the leaves was stronger than that of negatively-charged nanoparticles, which was mainly due to the electrostatic attraction between positively-charged nanoparticles and negatively-charged plant cell walls [26].

#### 2.2.3. Effect of Plant Species

Plant species are also one of the important factors that affect the absorption of nanoparticles in plant leaves [19]. The absorption of nanoparticles is related to the distribution, density and size of the pores in the leaves. For example, compared with dicotyledonous plants, the stomata of monocotyledonous plants are arranged neatly and have regular shapes. The growth stage and life cycle of plants also affects the absorption of nanoparticles in leaves. Most plant species only have stomata on the lower epidermis, while a few have stomata on both upper and lower epidermis [29,30,31]. When there were stomata on both sides of leaves, the number of stomata on the lower epidermis of dicotyledon plants was about 1.4 times higher than that on the upper epidermis. For monocotyledon plants, the number of stomata was similar on both sides [32]. In addition, abiotic environmental factors such as temperature, humidity and light also affect the opening and closing of stomata, thus affecting the absorption of NPs [33].

Studies showed that the absorption of CeO_2_ NPs in dicotyledonous pumpkin was more efficient than that in monocotyledonous wheat [34]. The absorption rate of Ce NPs in tomato was higher than that in *festuca* [35]. In recent years, Hu and his colleagues demonstrated that monocotyledonous plants represented by maize have smaller extracellular space in their leaves, which is not conducive to the entry of nanoparticles. While dicotyledon plants, represented by cotton, have higher stomatal density, which provides more opportunities for NPs to enter [5,18].

## 3. Absorption of Nanoparticles in Plant Roots

### 3.1. Pathways for Root Uptake

The contact between nanoparticles and plant roots initially occurs through adsorption on the root surface. Because the root hairs can release chemical substances such as mucus or organic acids, the root surface has negative charges, making nanoparticles with positive charges are more likely to accumulate in the root and be easily to be absorbed on the root surface [36,37]. The formation of lateral roots can create a new adsorption interface for nanoparticles, thus providing the possibility for nanoparticles to enter the root column [38] (Figure 1). The composition and function of the plant root epidermis are similar to that of plant leaf surface. However, the epidermis on plant root tip surface and root hair surface of the primary and secondary roots are not fully developed. When nanoparticles are exposed to this area, they directly contact the root epidermis and cross it [5]. The epidermal cells of the root cell wall are semi-permeable. The root cell wall contains small pores that can prevent the passage of large particles [13]. When the root lacks exodermis, nanoparticles can enter the central column or xylem of the root [39]. Other studies have shown that some nanoparticles can destroy the plasma membrane and induce the formation of new pores on the epidermal cell wall to facilitate the entry of large-diameter nanoparticles [40]. When nanoparticles enter plant tissue, they can be absorbed by plant cells through multiple pathways such as ion pathway, endocytosis, binding with cell membrane proteins or physical damage [36].

Some studies showed that nanoparticles were absorbed by plant roots and infiltrated into cells mainly through the hydrophilic pathway. However, due to the small pore size, the entry of nanoparticles into cells through this pathway is very limited [20]. Another important way for the absorption of NPs in plant cells is endocytosis. The invagination of plant cell plasma membrane brings NPs into cells. Studies have shown that plant protoplasts can internalize the particle with sizes less than 1 μm through endocytosis, thus, the nanoparticles absorbed by endocytosis have no particle size selectivity [41]. Liu et al. proposed that carbon-based nanoparticle and carbon nanotubes were absorbed by the root cells of *catharanthus roseus* through endocytosis [42]. In addition, nanoparticles can be absorbed by plants by combining with transport proteins on the outer epidermis [2].

### 3.2. Key Factors That Affecting the Uptake of Nanoparticles via Roots

Different factors have a significant impact on the absorption of nanoparticles by plant roots, including the size, chemical composition and surface charge of nanoparticles [43].

#### 3.2.1. Effect of Size and Chemical Composition

The size of nanoparticles is considered to be the most important factor affecting the root absorption of nanoparticles. Previous studies have shown that nanoparticles with a particle size less than 10 nm can be absorbed by plant roots, such as gold nanoparticles (3.5 nm) [44] and CeO_2_ nanoparticles (8 ± 1 nm) [45], respectively, in the roots of *Vicia faba* L. and maize. At the same time, other reports have confirmed that the absorption of nanoparticles by wheat roots is related to the size of NPs. For example, TiO_2_ nanoparticles with particle size of 36–140 nm can be absorbed by wheat roots. As the particle size increases, the total amount of absorption reduces. While TiO_2_ NPs with particle size greater than 140 nm cannot be absorbed [46]. Because of the size limitation, it is generally believed that if the particle size exceeds 100 nm, it is difficult for metal-based nanoparticles to be absorbed by plants through the roots [16].

However, it is interesting to find that silicon-based and natural polymer-based nanoparticles with particle sizes above 100 nm can be absorbed. For instance, Si NPs with a particle size of 200 nm were absorbed by *Arabidopsis* root after 6 weeks of treatment [47]. Zein nanoparticles with an average particle size of 135 ± 3 nm were proven to be absorbed by sugarcane roots by using confocal microscopy and transmission electron microscopy [48].

#### 3.2.2. Effect of Surface Charge

In addition to its size, the uptake of NP by plant also depends on its surface charge [49]. The negative charge of plant root cell walls determines the surface charge properties of nanoparticles that can be absorbed by plant roots. The effect of electric charge on the absorption of nanoparticles in plant roots is slightly different from that in leaves [50]. Positively charged nanoparticles tend to accumulate on the root surface because of electrostatic attraction between the negatively charged of the cell wall and the positively charged nanoparticle, but cannot enter the root tissue [51].

O’Reilly and Napier et al. prepared nanoparticles with different particle sizes (20–100 nm) and different surface charges by using the method of reversible addition chain transfer-polymerization. By using confocal microscopy, they found that the uncharged nanoparticles and negatively charged nanoparticles (22 nm) could be absorbed by *Arabidopsis thaliana* root cells and then entered the xylem of the root. In comparison, the positively charged nanoparticles can only accumulate at the root epidermis and cannot enter the root tissue of *Arabidopsis* [52].

## 4. Transport and Distribution of Nanoparticles in Leaves

### 4.1. Transport Pathway of Nanoparticles in Leaves

When nanoparticles enter mesophyll cells through the cuticle and stomata, they can be transported for a long distance in plant through the extracellular pathway or the plastid pathway [53]. The extracellular pathway mainly refers to the transport through the extracellular space. When nanoparticles pass through the cell wall, they can be transported through the extracellular space (such as cell wall, a longitudinal channel between cell walls, intermediate lamella and xylem) according to their particle size and surface charge [54]. The protoplast pathway is mainly transported through intercellular channel, plasmodesmata, with a diameter of about 2–20 nm [55]. When nanoparticles pass through the plasmodesmata, they will accumulate in the cytoplasm and transport to the plant endothelial layer and Casparian strip [53] (Figure 1).

Some studies have shown that nanoparticles with particle size less than 50 nm are generally transported in plants through the plastid pathway, while most nanoparticles with particle size between 50 and 200 nm are transported through the apoplast pathway [56]. The subsequent transport of nanoparticles from leaf to root is then achieved by the vascular system–phloem transport pathways. Because the diameter of phloem sieve tube is large, the particle size about of 0.405 μm can also be easily transported through the phloem [17]. 

### 4.2. Distribution of Nanoparticles Absorbed via Leaves

A large number of studies have shown that nanoparticles applied through leaves can be transported to various parts of plants, such as stems, roots, flowers, fruits and even to the rhizosphere soil [57,58,59]. When CuO NPs were sprayed on maize leaves, nanoparticle deposits were observed at the epidermis of the leaves. After entering the cells, CuO NPs were transferred between cells through plasmodesmata [60]. In another study, after 48 h of spraying gold nanoparticles on watermelon leaves, the uptake and accumulation of gold nanoparticles in watermelon leaf cells were confirmed by ICP-MS. The gold was also detected in watermelon stems and roots, indicating that nanoparticles transported from leaves to roots through phloem [61]. Hong et al. also reported that after CeO_2_ nanoparticles acted on cucumber leaves for 72 h. CeO_2_ NPs were found in cucumber root slices by transmission electron microscopy, providing direct evidence for the transport of CeO_2_ NPs from leaves to cucumber stems and roots [62]. Bueno et al. prepared porous hollow silica nanoparticles loaded with azoline oxalate (Azo@PHSNs) and sprayed them on tomatoes. The content of Si in different parts of the plant was detected and quantified on the second and the fourth days after the action. The results showed that the content of Si in different tissues of tomato was stem > root > young leaf > mature leaf, which indicated that the transport efficiency of PHSNs to various tissues in tomato was different. Compared with the transport between different leaves, the transport of nanoparticles from stem to root was easier [63].

### 4.3. Key Factors That Affect the Transport of Nanoparticles Absorbed via Leaves

#### 4.3.1. Effect of Size and Chemical Composition

Previous studies showed that metal-based nanoparticles that could be transported to other parts of the plant had a smaller size, in comparison with nanoparticles made of nonmetal materials. This is because metal-based nanoparticles of a larger size cannot be absorbed. The metal-based nanoparticles that could be found in phloem or other parts of plant leaves had sizes of 30–50 nm [64]. While a larger nanoparticle made of non-metal materials was often found to be translocated in plant leaves [65].

Zhao et al. treated cucumber leaves with FITC-labeled mesoporous silica nanoparticles (MSNs) (particle size 200–300 nm) and found FITC-labeled MSNs in petioles, stems and untreated leaves and flowers after only 4 h, proving that MSNs can be translocated from leaves to other tissues of cucumber [66]. In the study by Bueno et al., the average particle size of the nanoparticles was 253 ± 73 nm. It was found that although the porous hollow silica nanoparticles (PHSNs) were much larger than the known size exclusion limit in plant tissues, PHSNs can not only be absorbed by leaves, but also be transported to stems, roots and young leaves [67].

Abdel-Aziz et al. reported the existence of chitosan NPK NPs (10, 25, 100 nm) in wheat phloem tissue with the help of a transmission electron microscope, which means that NPs are transported from leaves to stems, and then to roots through the phloem route [68].

#### 4.3.2. Effect of Surface Charge

The surface charge of nanoparticles will affect their transport in the extracellular pathway. The pectin on the cell wall has carboxyl groups that make the cell wall negatively charged, thus the positively-charged nanoparticles will be attracted by the negative charge on the cell wall, which makes the positively-charged nanoparticles easier to transport through the extracellular pathway [69].

Sun et al. sprayed the leaves of maize seedlings with carboxyl and amino-modified polystyrene nanoparticles and found that two kinds of PS NPs with different charges could effectively accumulate on the leaves of maize and the particles gathered at the stomata. Among them, the accumulation of amino-modified positively charged PS-NH_2_ nanoparticles in the leaves is significantly greater than that of carboxyl-modified negatively-charged PS-COOH nanoparticles. When the nanoparticles enter the leaves, the negatively-charged PS-COOH NPs can be more effectively transferred to the vascular system of plants and transferred to the roots through vascular bundles [25].

## 5. Transport of Nanoparticles at the Root

### 5.1. Transport Pathway

Compared with the application of different nanoparticles in plant leaves, there are more studies on the effect of nanoparticles on plant roots. When the nanoparticles act on the plant root, they are absorbed by the plant root hair cells. They selectively pass through the cell wall, enter the endothelial layer from the epidermis through the way of symplast or exoplast, then transport to the aboveground part through the xylem vessels [70]. From the inside to the outside, the plant root is successively composed of a pericycle and vascular column, cortex and epidermis. The vascular column includes a pericycle and vascular tissue, which is located at the center of the plant root [71]. The cortex consists of the endothelium and the exodermis, in which the endothelium is connected with the pericycle (Figure 1).

The transport of nanoparticles in the root is also carried out in two ways, namely, the exoplast and the eutectic. In the exoplast pathway, when nanoparticles are firstly transported through the epidermis to the endothelium, then they will be blocked by the Casparian strip. Some nanoparticles will deposit in the endothelium, and another part of the nanoparticles will be transported radially along the Casparian strip, but will not cross the Casparian strip. However, at the lateral root junction, the Casparian strip is not fully developed. Nanoparticles can reach the vascular system and transport through this area [72]. In the symplast pathway, NPs enter the cytoplasm through the plasma membrane and are transported between cells through plasmodesmata.

### 5.2. Key Factors That Affect the Transport of Nanoparticles Absorbed via Roots

#### 5.2.1. Effect of Chemical Composition

Previous studies have shown that small particle size nanoparticles such as TiO_2_ NPs, Cu NPs, CuO NPs, etc. can be absorbed by the roots of cucumber and corn, and be transferred to the aboveground part of plants. At the same time, nanoparticles can also be detected in leaves, fruits and other parts [60,73,74].

Chen et al. treated rice seedlings with 100 mg/L ZnO nanoparticles (40 nm) for 7 days, collected rice root tips, and observed the absorption and distribution of ZnO NPs in the root of rice seedlings through an ultrathin section and TEM. It was found that zinc oxide nanoparticles were found in the cell wall of rice root elongation zone and in the cytoplasm of cortex cells, indicating that ZnO NPs can penetrate the cell wall and aggregate in the cytoplasm of cortex cells [75].

Tong et al. prepared nanoparticles by coating metolachlor with monomethyl ether polyethylene glycol-polylactic acid-glycolic acid copolymer (mPEG-PLGA) with a size of about 100 nm, and studied its transport and distribution in rice by fluorescence labeling with Cy5. The experimental results showed that there was an obvious fluorescence signal in the roots of rice. The fluorescence-labeled mPEG-PLGANPs may internalize into the rice plant through the extracellular pathway. In another study, PS NPs with a size of 200 nm acted on the root of wheat. Two hours later, under the fluorescence microscope, fluorescence signals were found on the outside of the root epidermal cell wall [76]. Because the Casparian strip at the root meristem of wheat was not mature, NPs entered the vascular system through the root apical meristem. After two hours of treatment, they could be found in the vascular system of the xylem, which proved that nanoparticles were transported through the extracellular pathway [20].

#### 5.2.2. Effect of Surface Charge

When nanoparticles enter the plant roots, the effect of surface charge on the transport of nanoparticles is the same as that of the transport of nanoparticles through leaves. In plant roots, the roots secrete negatively-charged mucus and secretions, which become the first barrier for nanoparticles to enter the plant roots, and inhibit the absorption of positively-charged nanoparticles on the outside of the cell wall. The plant cell wall has negative charges, thus the positively-charged nanoparticles are more likely to accumulate at the cell wall after being absorbed. While the negatively-charged nanoparticles will transport in the root and the transport efficiency of the electrically neutral nanoparticles in the root is more obvious [52]. Sun et al. used CeO_2_ nanoparticles with different charges to treat the root of wheat and found that among the nanoparticles adsorbed on the root surface, CeO_2_ NPs with positive charges were significantly more than those with negative charges, while CeO_2_ NPs with negative charges could be transferred to different parts [50]. Avellan et al. obtained similar results in the study of absorption and transport of gold nanoparticles with different charges in *Arabidopsis* root [77].

## 6. Effect of Nanoparticles on Plant Physiological Activity

Nowadays, due to the advantages of nanoparticles as nano-carriers and nano-pesticides, such as their small size, ease of use, capacity for long-term storage, and improvement of the efficiency of agricultural chemicals, the application of nanoparticles in agriculture is considerably more extensive [63]. The interaction between nanoparticles and plants has beneficial and harmful effects on the plant’s physiological morphology, plant development and yield of crops. The effects of nanoparticles on plants are related to plant species, use methods, dosage and concentration of nanoparticles.

### 6.1. Nanoparticles Promote Plant Development and Yield

The use of agricultural nanoparticles has improved the quality of plant products better than traditional pesticides, which has been proven by many researchers. Nanoparticles play an important role in plant growth and improving plant quality by increasing nutrient content, improving photosynthetic activity and metabolism [78] (Figure 2). Zinc oxide nanoparticles have been confirmed to participate in the synthesis and photosynthesis of plant chlorophyll and the formation of starch, thus increasing the concentration of soluble carbohydrates [79]. The use of ZnO nanoparticles can improve the antioxidant activity and chlorophyll content of cotton, increase the number and weight of cotton bolls per plant, and improve cotton fiber quality parameters such as uniformity and fiber strength [80]. ZnO NPs act on tomato plants and improve tomato yield by increasing the absorption of nutrients (phosphorus and zinc) [81]. Fe_3_O_4_ NPs can improve plant biomass and productivity by increasing the content of protein, nutrients, and carbohydrates in plants [81,82]. Sharifi et al. used Fe_3_O_4_ NPs to act on corn plants, and Armin et al. used Fe_3_O_4_ NPs to act on wheat plants. All their results confirmed the above conclusions [83,84]. In addition, when Fe_3_O_4_ NPs were used to treat wheat at the tillering stage, the number of wheat seeds increased significantly. Gao et al. found that TiO_2_ NP increased the biomass accumulation of spinach by 60% [85]. In addition, CeO_2_ NPs promoted stem elongation at 1–10 mg/L, and fruit weight significantly increased at 10 mg/L [86].

As an efficient environment-friendly photocatalyst, TiO_2_ nanoparticles have been proven to be able to improve light absorption by improving the energy conversion of the light system, and have antibacterial activity after surface chemical modification, which can reduce the half-life of pesticides and promote seed germination and seedling growth [87]. In 2013, it was proven that TiO_2_ nanoparticles could improve the photosynthetic efficiency of spinach. In recent studies, it was shown that TiO_2_ NPs could promote wheat growth and increase wheat yield. At the same time, experiments have proven that these improvements are due to the fact that titanium dioxide nanoparticles promote cycling and linear phosphorylation to improve photosynthetic activity, thus increasing the supply of photosynthates and in turn increasing plant yield. Spraying TiO_2_ NPs on the leaves improves the dry matter yield of barley because nanoparticles improve the photoreduction activity [88]. In addition, under the joint action of ZnO NPs and Si NPs, the yield, weight and sugar content of mango fruit increased [89]. The promotion effects of different kinds of nanoparticles on plant growth and seed germination are listed in Table 1.

### 6.2. Nanoparticles Alleviate Plant Abiotic Stress

Plant abiotic stress is the main problem that plants face in the process of growth due to drought, salt and heavy metal elements, which lead to the reduction of crop yield due to plant growth retardation. In order to alleviate these stresses, plants have evolved different defense mechanisms through physiological pathways. The application of nanoparticles can help plants to alleviate abiotic stresses. Due to environmental, climatic and other factors, the yield and biomass of crops in arid areas are significantly reduced compared with a normal environment. Sun et al. applied zinc oxide nanoparticles to treat corn and found that nanoparticles can improve the photosynthetic rate and chlorophyll content of plants under drought stress, which proved that nanoparticles have a mitigating effect on plant drought stress [59].

Inappropriate salinity in plant growth environments will lead to plant nutrition imbalance and slow plant growth [99]. The application of nanoparticles in agriculture can help improve the activity of enzymes involved in the salt tolerance mechanism in plants. The study showed that ZnO NPs alleviated the salt stress of cotton [100] and wheat [101]. The application of SiO_2_ NPs on the leaves increased the elasticity and expansion of the cell wall of cucumber during the growth period and increased the accumulation of nitrogen and phosphorus elements in the leaves by reducing the loss of leaching process, thus reducing the content of Na, alleviated the salt stress on cucumber plants [66].

The existence of heavy metals is more harmful to plant growth, which will affect plant morphology, inhibit plant growth and stimulate plants to produce oxidative stress. In order to resist heavy metal stress, nanoparticles can improve the ability of antioxidant systems by regulating the concentration of heavy metal ions in soil, reducing the expression of heavy metal ion transport genes in plants, stimulate the production of defense substances (such as organic acids, root exudates and phytochelatin) to cope with the stress of heavy metal ions in plants [102].

Sharifan et al. used ZnO NPs and heavy metals Pb^2+^ and Cd^2+^ as the hydroponic culture system to culture lettuce (*Lactuca sativa* L. var. *Longifolia*), and detected the content of Pb and Cd in plant tissue. It was found that ZnO NPs significantly reduced the accumulation of Pb and Cd in the root of lettuce, which were 81% and 49%, respectively. Pb and Cd in the shoots of lettuce decreased by 44% and 30% [103], respectively. Yan et al. treated rice with ZnO NPs, proving that zinc oxide nanoparticles can improve rice growth and reduce the accumulation of arsenic in rice. The best effect was found when the concentration of ZnO NPs was 100mg/L. Compared with the control group, the concentration of As in the bud and root of rice seedlings decreased by 40.7% and 31.6%, respectively. Se and Si NPs reduced the absorption and accumulation of Cd and Pb in rice, thus reducing the lethal effect of heavy metals on plants. At the same time, the use of SiO_2_ NPs in leaves helps to increase chlorophyll content and reduce the accumulation of Cd in rice [104].

### 6.3. Toxicity of Nanoparticles to Plants

In addition to many beneficial effects, the toxicity of nanoparticles in plants cannot be ignored. The toxicity of nanoparticles can damage plants in a variety of ways, for example, by stimulating plant oxidative stress, resulting in physical damage to plants, such as stomatal closure due to the aggregation of nanoparticles [6].

#### 6.3.1. Inhibitory Effect of Nanoparticles on Plant Growth

When nanoparticles are sprayed on plants through leaves at high concentrations, a large number of NPs will gather on the surface of leaves, which results in stomatal blockage and hinders the gas exchange and photosynthesis of plants (Figure 2). Some studies have shown that when the concentration of Zn NPs and Cu NPs exceeds the critical value, the plant growth rate will slow down and the leaves will turn yellow.

CuO NPs are toxic to *H. sativum*, and will reduce the photosynthetic rate of plants and inhibit the growth of roots and stems [105]. At the same time, nanoparticles transform in plants result in damage to cell structure and reduce the absorption and transportation of nutrients [106].

For polymer-based nanoparticles, the current research mainly discusses their toxic effects on plants in the process of application. For example, after treating the seeds of *Lepidiumsativum* with plastic particles with a size of 50 and 500 nm for 8 h, the germination rates decreased by 56% and 46%, respectively [51]. When ryegrass was exposed to PLA nanoparticles for 30 days, the germination rate decreased by 6% [107]. In contrast, the germination rate of wheat seeds and onions treated with PS NPs for 72 h had no effect [107,108]. Therefore, long-term exposure of plants to polymer nanoparticles will reduce the germination rate, which is mainly due to the reduction of nutrient absorption of plants due to the blockage of pores by nanoparticles.

In addition to the effects of polymer nanoparticles on plant seed germination, other studies have reported the effects on plant seedling growth. The length of the onion root decreased by 41.5% after polystyrene nanoparticles (50 nm) acted on the onion root for 72 h [109,110]. The length of the *Lemna* minor root also decreased under the effect of polyethylene nanoparticles. In *Arabidopsis thaliana* treated with polystyrene nanoparticles with different surface charges (PS-NH_2_, PS-SO_3_H), the fresh weight of *Arabidopsis thaliana* decreased by 50% on average. The plant length decreased by 15%, the root length decreased by 30% on average, and the higher the concentration of nanoparticles, the more obvious the inhibition of *Arabidopsis thaliana* seedlings is [49]. A better understanding of the effects of nanoparticles on plants can help assess their toxicity (Table 2).

#### 6.3.2. Genotoxicity and Oxidative Stress Damage of Nanoparticles in Plants

The transport and accumulation of nanoparticles in plants will induce phytotoxicity and the interaction between nanoparticles and plants will lead to increased production of plant ROS, resulting in oxidative damage and genetic toxicity (Figure 2) [129,130]. Similarly to all aerobic organisms, plant cells activate ROS in response to environmental changes [131]. When ROS levels exceed defense mechanisms, cells are placed in a state of “oxidative stress”, causing unlimited damage to proteins, nucleic acids and lipids in cell membranes and inducing oxidative stress [132]. Plants can protect cells from the toxicity of reactive oxygen species (ROS) through antioxidant enzymes and antioxidants. Plants can protect cells and subcellular systems from the toxicity of active oxygen radicals by antioxidant enzymes and low molecular weight antioxidants [133]. Therefore, the research on oxidative damage caused by nanoparticles mainly focuses on the determination of antioxidant enzyme activity and ROS content. Some research and analysis, especially when nanoparticles act on plants at high concentrations, results in excessive accumulation of nanoparticles in plants and a large number of ROS that activate the antioxidant system [58,106,134]. As the concentration of plastic NPs increased, the activity of several representative antioxidant enzymes in rice roots increased, suggesting that the plant could stimulate a defensive response and remove the excessive accumulation of ROS [135]. Meanwhile, a higher concentration of nanoparticles will affect the level of plant endogenous hormones [58]. At present, there are also corresponding studies to evaluate the impact of nanoparticles on plant endogenous hormones [136]. For example, iron oxide nanoparticles with a concentration of 100 mg/L reduced the yield of Bt-transgenic cotton and led to an increase in hormone levels. While in the root of Bt-transgenic cotton, the hormone content decreased [137]. In addition, metal-based nanoparticles transform and release toxic metal ions in plants, which destroy DNA and protein in plants and inhibit normal cell metabolism [138].

In addition to inducing ROS production in plants and triggering hormonal changes in plants, nanoparticles are also genotoxic. Nanoparticles interact with biological macromolecules such as nuclei and lipids and exert genotoxicity, which can affect plant cell division [139]. For example, Ag NPs internalized in wheat root tips have been shown to interfere with normal cell division, inhibit DNA synthesis and lead to chromosome aberrations [140]. In addition, the mitotic index of onion cells exposed to PS NPs was significantly reduced and chromosomal aberrations and nuclear abnormalities were observed, leading to the destruction of genomic stability, demonstrating the genotoxicity of PS NPs [141].

In order to deal with the toxicity caused by nanoparticles, researchers have also developed corresponding solutions, in which the toxic substances released by nanoparticles can be effectively reduced by adding a surface coating. For example, adding an iron coating to the surface of zinc oxide nanoparticles can effectively reduce the release of zinc ions. Fe-coated zinc oxide nanoparticles have no inhibition on plant germination and pigment content in plants [142]. The method of reducing toxicity through encapsulation or surface modification of nanoparticles is also applicable to polymer nanoparticles [43].

Analyzing the toxicity of different nanoparticles as agricultural chemicals are helpful to determine the optimal concentration of nanoparticles in plant growth. As nontoxic, degradable and biocompatible compound nanoparticles, natural polymer nanoparticles can largely avoid the negative effects of metal-based, silicon-based and organic-based nanoparticles on plants and the environment [143]. Chitosan is the only positively-charged polysaccharide in nature, which has been reported as a nanoparticle material due to its antibacterial and antiviral properties. Because of its positive charge, chitosan nanoparticles enhance the affinity with the plant cell membranes and increase the reactivity of the plant systems. When chitosan-based nanoparticles are used as plant growth promoters to treat seeds and seedlings; it has been proven that they can improve plant nutrient absorption, chlorophyll content and photosynthesis rate. Three different sizes of chitosan nanoparticles (420 nm, 750 nm, 970 nm) were applied to plants. The results showed that plant chlorophyll increased by 61%, 81% and 61% and the photosynthetic rate increased by 29%, 59% and 72%, respectively [144]. The application of chitosan nanoparticles on wheat and barley has also been proven to promote plant growth [145].

## 7. Conclusions and Prospectives

This article reviews the absorption and transportation of nanoparticles in plants, as well as the impact of nanoparticles on plant production. Nanoparticles are absorbed by plant leaves through the stratum corneum pores and stomata of the leaves and absorbed by the roots through the primary roots, root cell wall pores and damaged areas. These nanoparticles are transported within the plant body through both the symplastid and apoplastic pathways and transported between different tissues of the plant through the xylem and phloem. The physical properties of nanoparticles such as size, chemical composition, surface charge and surface modification determine the absorption and transportation process. In addition, nanoparticles have different effects on plant seed germination and plant growth during transportation within the plant body. It has been proven that various nanoparticles can promote plant growth at low concentrations, while causing plant damage at high concentrations. The toxic effect of nanoparticles will also lead to changes in plant hormone levels and even lead to genotoxicity, such as chromosome aberration and micronucleus formation, thus changing gene expression in plants. Although the impact of nanoparticles on plants has been widely confirmed, the mechanism of plant toxicity induced by nanoparticles is still limited.

Meanwhile, the use of natural polymer materials has emerged as a preferred method for manufacturing nanoparticles due to their biodegradable and edible properties, surface-modifiability and ability to synthesize functional nanoparticles by design. However, the uptake and translocation of natural polymer nanoparticles in plants are still relatively unexplored and there is much work to be done in improving their production and application.

Furthermore, the traditional quantitative methods for analyzing nanoparticles are not always applicable to natural polymer nanoparticles and appropriate methods for accurate quantitative analysis are yet to be proposed.

Finally, in the actual agricultural production process, attention should be paid to the effect of nanoparticles and other substances in the environment on the stability and morphology of nanoparticles under natural conditions. The use of bio-based natural polymer nanoparticles as an emerging technology to improve agricultural production should be further explored, so as to optimize the interaction of specific species in a specific environment according to the used nanoparticles, concentration and time, so that these technologies can help the expected plant production and cultivation.

## Figures and Tables

**Figure 1 materials-16-03097-f001:**
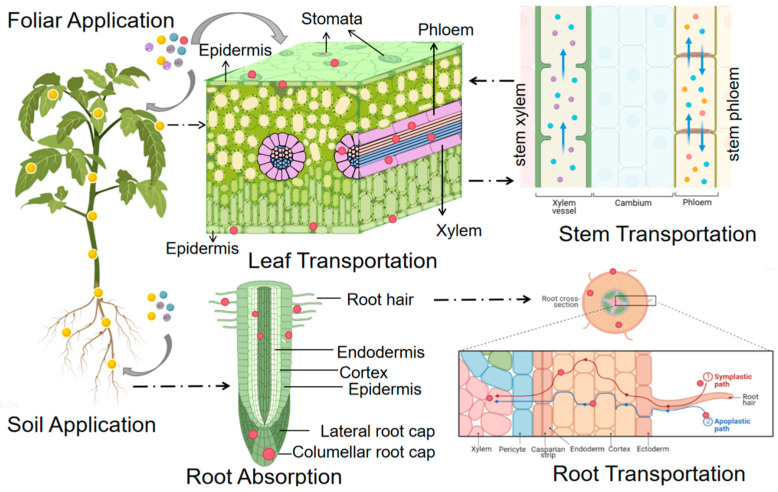
A schematic diagram of the uptake and translocation of NPs in plants through foliar application or root exposure treatment.

**Figure 2 materials-16-03097-f002:**
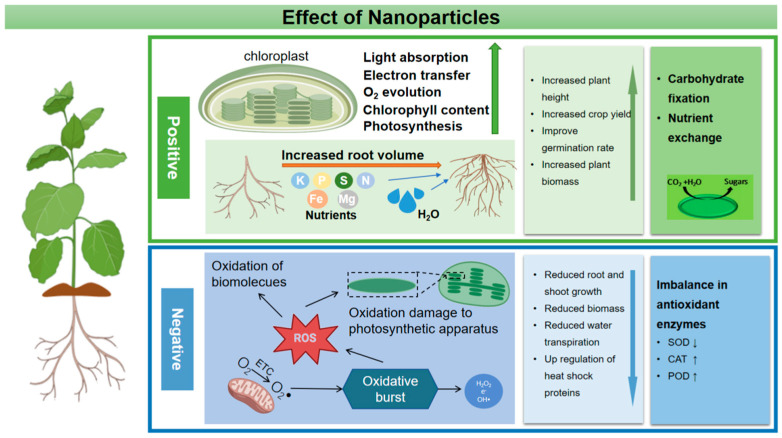
A schematic diagram of the physiological activity of NPs in plants.

**Table 1 materials-16-03097-t001:** Summary of the representative of nanoparticles that promote plant development and yield.

Nanoparticle	Size (nm)	Concentration	Plants	Positive Effect	Reference
ZnO	18	Fresh soil with 6 mg/kg soil	*Triticum aestivum Landmark*	Leaf chlorophyll levels and shoot height increased	[90]
	30	0 and 500 mg/kg soil with and without organic P (0, 20 and 50 mg/kg)	*Maize*	ZnO NMs increased root dry weight	[91]
	<100	50, 100 and 150 mg/L	*Mangifera indica Linn*	Improved the total yield (fruit number and weight per tree) the combined application of NPs resulted in an increase in fruit yield, average fruit weight, length, width, TSS, sugars and displayed the lowest acidity percentage	[89]
	12–24	1000 and 2000 mg/L	*Capsicum chinense*	NPs promoted plant growth, increased number and average weight of the fruits, fruit quality, capsaicin and dihydrocapsaicin content at low doses	[92]
TiO_2_	20	25–750 mg/L	*Oryza sativa*	Enhanced accumulation of palmitic acid, amino acids and glycerol in rice grain, improved shoot growth and phosphorus concentration in whole plant and grains	[93]
	32	10, 100,1000 mg/L	*Triticum aseivum*	Enhanced growth of lateral roots and biomass with concurrent uptake of titanium	[94]
	27	250, 500, 750 mg/L	*Cannabls sativa*	Increased potassium and phosporus in cucumber fruits	[73]
Cu	50	50–500 mg/L	*Solanum lycopersicum*	Enhanced lycopene, vitamin-C in tomato fruits, number of fruits and fruit firmness	[95]
SiO_2_	4–10	5 mM	*Oryza sativa*	Increased grain yield and weight	[96]
Si		25, 50 mg/L	*lentil*	Promote seed germination, seedling vigor and biomass	[97]
CeO_2_	15–30	200, 500 mg/L	*Arabidopsis thaliana*	Increased the root elongation, root and shoot growth	[98]

**Table 2 materials-16-03097-t002:** Summary of the inhibitory effects of nanoparticles on plants.

Nanoparticle	Size (nm)	Concentration	Plants	Negative Impact	Reference
ZnO	<50	500 ppm	*Glycine max*	Inhibition of root elongation, cell viability and biomass, generation of superoxides, reduced biomass of foliage, alteration of gene expression	[111]
	90	400–3200 mg/kg	*Zea mays*	Increased production of superoxide anions and superoxide dismutase activity decreased mineral nutrient acquisition, decreased photosynthesis and root activity	[112]
	30–40	0.02–2 g/L	*Zea mays*	Negative effect on seed germination and seedling growth	[113]
	10	250, 500, 750 mg/kg	*Medicago sativa*	Reduced root biomass up to 80%	[114]
	64, 80	20–200 mg/L	*Vigna angularis*	Disrupted plant physiology of plant, enhanced oxidative stress and reduced photosynthetic pigment	[115]
Ag	10	10–50 mg/kg	*Lysopersicon esculentum*	Induced reactive oxidative stress that reduced photosynthesis, CO_2_ assimilation and fruit yield	[116]
	10	0.001–10,000 mg/L	*Allium cepa*	Root growth was inhibited	[117]
	10	25, 50, 75 and 100 μM	*Allium cepa*	Strongly reduced the root growth, induced mitotic index, induced ROS formation, caused oxidative DNA damage in higher concentrations	[118]
	5–10	300–900 ppm	*Lupinus termis*	Decreased germination percentage, reduced the root and shoot elongation, root and shoot fresh weights, total chlorophyll, total protein content and sugar content	[119]
CuO	40–80	500, 1000, 2000 mg/L	*Zea mays, Oryza sativa*	95% and 97% inhibition in root length of maize and rice at 2000 mgL^−1^	[120]
	25–55	50–500 mg/L	*Brassica rapa*	Synthesis of photosynthetic pigments and sugar was decreased	[121]
	43	50–1000 mg/kg	*Oryza sativa*	Low water uptake by root and aerial parts; grain production was considerably reduced	[122]
TiO_2_	20 ± 5	50, 200 mg/L	*Oryza sativa*	Reduced grain yield and biomass	[123]
	25	250–1000 mg/L	*Cannabls sativa, Brassica oleracea var. capitata, Avena* *sativa, Zea mays, Lactuca sativa, Allium cepa*	Inhibition of root growth for oat, corn, cabbage, lettuce and reduction of soybean and cucumber germination	[124]
	5–15	0.02–2 g/L	*Zea mays*	Inhibition of germination, root and shoot growth	[113]
	20	100–500 mg/L	*Oryza sativa*	Reduced biomass and altered antioxidant defense	[125]
Al_2_O_3_	<50	10–1000 mg/L	*Sinapis alba*	At all concentrations, seed germination was affected negatively	[126]
	13	5, 25, 59 mg/L	*Triticum aseivum*	H_2_O_2_ content was increased, reduced production of photosynthetic pigment and anthocyanin	[127]
Au	2–21	5, 8, 10 mg/L	*Hordeum vulgare*	Leaves became yellow and necrotic, the roots colored dark brown and decreased fresh plant biomass, leaf and root lengths	[128]
	3.5, 18	48 ppm	*Nicotiana xanthi*	Caused biotoxicity and observed leaf necrotic lesions	[6]

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
