# Peer review of "Nanoparticles in Plants: Uptake, Transport and Physiological Activity in Leaf and Root"

_materials, 2023, doi:10.3390/ma16083097_

Round 1

Reviewer 1 Report

Dear Authors,

The present manuscript titled "Nanoparticles in plants: uptake, transport and physiological activity in leaf and root" talks about the role of NPs on various aspects of plant development. There are few suggestions that needs to be addressed for the quality improvement of the manuscript.

All the suggestions/comments are made in the pdf file.

The introduction part needs to be rewritten.

Most of the places citation is missing.

It will be good if the authors consider adding information on the biochemical aspects as well.

Reviewer 2 Report

Dear authors,

The author should provide the suitable references in the section 1. Introduction.

Old references should be replaced.

The author should carefully revised this manuscript. It is found many similarity sentences.

Overall manuscript related content and more recent references needs to be provide.

Typographical errors must be corrected throughout the manuscript (i.e, superfluous spaces, inconsistent use of units, superscript, etc.).

Reviewer 3 Report

In this review proposal, Wang and colleagues address the state-of-the-art on nanoparticle technology in plants. In general, the structure of the manuscript is well crafted and the "take-home" messages are quite clear. That being said, I would like to stress out that the authors are missing a few key points that should be included before further consideration of their review proposal. Specifically, the authors are mostly including the positive effects of nanoparticles on plants, while the potential deleterious effects on plants have less attention. Perhaps adding a "Table 2" would nicely fit in this purpose. In addition, a new section on the known persistence of nanoparticles in plant herbivores or plant produce consumers in general would clarify whether current nano particle applications in plants are harmless to the environment or not. 

Minor edits are required throughout the manuscript where appropriate (i.e. italics for plant species "Arabidopsis"). On Fig. 2 Change "Increase" for "Increased" and homogenize font size throughout the manuscript (i.e. First paragraph on the Introduction Section).
